# Taming generative video models
# for zero-shot optical flow extraction

**Seungwoo Kim**[*] **Khai Loong Aw**[*] **Klemen Kotar**[*]
**Cristobal Eyzaguirre** **Wanhee Lee** **Yunong Liu** **Jared Watrous** **Stefan Stojanov**
**Juan Carlos Niebles** **Jiajun Wu** **Daniel L.K. Yamins**

Stanford University

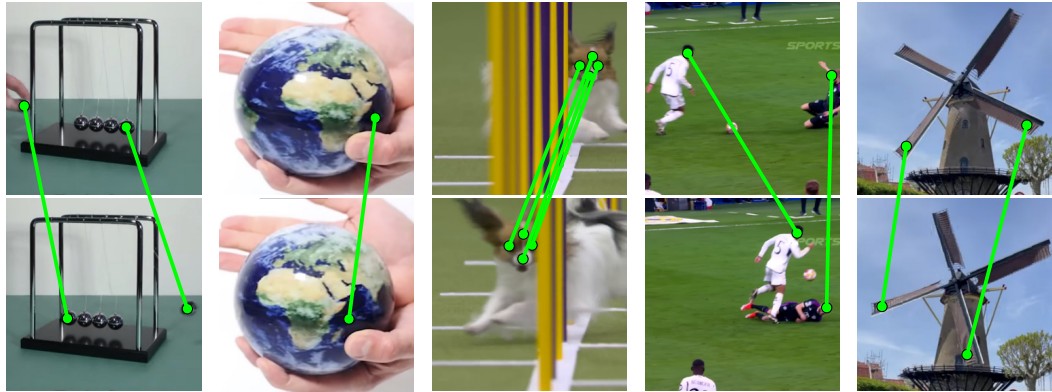

Figure 1: **We introduce a zero-shot test-time inference procedure called *KL-tracing*, which extracts robust optical flow and point tracking from a generative world model on challenging *in-the-wild* videos.** In every column, the green line links the query location in the first frame (top) to the position predicted by our method in the second frame (bottom). All clips are real-world internet videos and contain phenomena that classical, appearance-based optical flow methods find challenging: (A) Newton's cradle, where both frames have four balls in the middle, but the balls are different; the example involves physical reasoning. (B) Globe has challenging in-place object rotation and the query point is in the textureless ocean. (C) Dog weaving through occluding poles with large, rapid motion, including depth changes and motion blur. (D) Soccer tackle with fast, diagonal motion with motion blur and partial occlusion. (E) Windmill rotation where the repetitive blades and uniform sky make local matching challenging. These examples highlight the benefits of leveraging a powerful world model to extract optical flow for challenging real-world scene dynamics.

## Abstract

Extracting optical flow from videos remains a core computer vision problem. Motivated by the recent success of large general-purpose models, we ask whether frozen self-supervised video models trained only to predict future frames can be prompted, without fine-tuning, to output flow. Prior attempts to read out depth or illumination from video generators required fine-tuning; that strategy is ill-suited for flow, where labeled data is scarce and synthetic datasets suffer from a sim-to-real gap. Inspired by the Counterfactual World Model (CWM) paradigm, which can obtain point-wise correspondences by injecting a small tracer perturbation into a next-frame predictor and tracking its propagation, we extend this idea to generative video models for zero-shot flow extraction. We explore several popular architectures and find that successful zero-shot flow extraction in this manner is

---
[*]Equal contribution

39th Conference on Neural Information Processing Systems (NeurIPS 2025).

aided by three model properties: (1) distributional prediction of future frames (avoiding blurry or noisy outputs); (2) factorized latents that treat each spatio-temporal patch independently; and (3) random-access decoding that can condition on any subset of future pixels. These properties are uniquely present in the recently introduced Local Random Access Sequence (LRAS) architecture. Building on LRAS, we propose KL-tracing: a novel test-time inference procedure that injects a localized perturbation into the first frame, rolls out the model one step, and computes the Kullback–Leibler divergence between perturbed and unperturbed predictive distributions. Without any flow-specific fine-tuning, our method is competitive with state-of-the-art, task-specific models on the real-world TAP-Vid DAVIS benchmark and the synthetic TAP-Vid Kubric. Our results show that counterfactual prompting of controllable generative video models is an effective alternative to supervised or photometric-loss methods for high-quality flow. [1]

# 1   Introduction

Extracting motion information (optical flow) from videos is a fundamental yet open challenge in computer vision with many applications. Due to the intractable cost of obtaining ground-truth labels from real-world videos, most supervised baselines [39, 41] are trained on synthetic datasets, e.g., FlyingChairs [12], FlyingThings [28] and Sintel [6]. While invaluable for progress, these datasets cover a narrow slice of motion statistics–predominantly rigid objects, limited lighting variation, and short temporal horizons. As a result, they under-represent the long-tail of real-world phenomena such as non-rigid deformation, atmospheric effects, rapid camera shake and textureless regions. Self-supervised models [22, 37] that can be trained on real videos attempt to bridge this gap, but they often rely on task-specific heuristics, such as photometric consistency or smoothness which fail under complex lighting, occlusion, or long-range motion. Consequently, both supervised and self-supervised optical flow baselines struggle to generalize to challenging in-the-wild videos.

Inspired by successes across vision and language where large general-purpose models outperform smaller task-specific ones, we explore large-scale video models as a possible solution. Trained on massive repositories of real-world data, modern video models already demonstrate strong scene understanding [5, 30, 1, 27], suggesting an implicit grasp of optical flow [3, 40]. However, prior work extracting visual intermediates such as depth and illumination from these models still required supervised fine-tuning [34, 11]. Extending that recipe to optical flow would again depend on synthetic labels, facing the same sim-to-real domain gap. This motivates our search for a zero-shot procedure that can extract accurate optical flow from off-the-shelf video models without any additional training.

In fact, such a procedure has been proposed in the Counterfactual World Model (CWM) framework [3]. Zero-shot optical flow is obtained by adding a small "tracer" perturbation to the source frame and tracking how a pretrained next-frame predictor propagates the tracer to the target frame (Section 3). In practice, however, because deterministic video models like CWM can only predict a single future state, it encourages predictions that average over future possibilities, yielding perceptually blurry frames that wash out the injected tracer, leading to less precise motion estimates (Section 4.1).

To overcome this, we implement the perturb-and-track method with *generative* video models, which generate crisp predictions as they sample from a distribution instead of regressing to a mean. This is insufficient, however, as each state-of-the-art generative video model faces its own flow extraction challenges. Stable Video Diffusion (SVD) [5] produces photorealistic frames, but its conditioning is weak and non-localized: generation is guided by a single global latent inverted from the target frame, so pixel-level edits introduce noisy differences, corrupting the extracted flow (Section 4.2). The recent Cosmos model [30] can condition generation on ground-truth patches from the target frame, but its autoregressive rollout is strictly in raster order, so these provided patches must all lie in the raster top-left region of the image, far less informative than the same number of patches distributed randomly across the image, thus failing to accurately reconstruct the target frame (Section 4.3).

Through a systematic analysis of state-of-the-art video models, we find three key properties for accurate zero-shot optical flow extraction: (1) distributional prediction of future frames (avoiding blurry or noisy outputs); (2) local tokenization, which encodes each spatio-temporal patch independently; and

---

[1]Project website at: `https://neuroailab.github.io/projects/kl_tracing/`

(3) random access decoding that can condition on any subset of future pixels. These are present in the recent Local Random Access Sequence (LRAS) architecture [25], whose patch-level conditioning and random-order decoding provide much more fine-grained controllability than previous generative video models (Section 4.4). In addition, we propose a new method, *KL-tracing*, that further improves upon the probe-and-track method by using the autoregressive nature of LRAS which exposes the probability distributions for predictions. KL-tracing computes the perturbation difference in the prediction logit space, thereby bypassing noisy RGB differences resulting from sampling randomness. We achieve state-of-the-art results on the challenging real-world TAP-Vid DAVIS dataset and the synthetic TAP-Vid Kubric dataset relative to supervised and unsupervised flow baselines (Section 6).

Overall, we show how to *tame* generative models—unruly, hard to control, and normally only steered with previous frames or coarse text prompts—into a tool that obeys fine-grained patch conditioning and KL-tracing for extraction of precise optical flow. Our contributions are threefold: (1) We provide the first systematic study of optical flow extraction from large generative video models, highlighting failure modes of both deterministic regressors and underconstrained generative models, and discover key model properties for flow extraction; (2) we introduce *KL-tracing*, a simple yet effective test-time inference flow procedure for tracing perturbations through the predicted distributions of future states; and (3) we tame LRAS with KL-tracing to obtain both quantitative and qualitative results of accurate flow traces on both real-world and synthetic benchmarks, as well as in-the-wild videos.

## 2 Related Work

**Optical Flow** refers to estimating the per-pixel 2D motion between a pair of video frames. Most models [39, 41] are supervised on synthetic video datasets [12, 28, 6], posing a sim-to-real gap, i.e., predominantly rigid objects, limited lighting variation, and short temporal horizons, failing to capture the long tail of real-world phenomena such as non-rigid deformation, atmospheric effects, fast camera shakes, and fine-scale textureless regions. Alternatively, self-supervised models [23, 26, 37] train on real-world videos, but often rely on task-specific architectures and heuristics such as photometric consistency and smoothness loss [23, 26, 37]. Regardless, these optical flow methods are designed for short frame gaps and relatively non-complex scenes with little motion dynamics.

**Point Tracking** follows a set of points across longer time horizons. Most methods are supervised [9, 17, 24], often on synthetic datasets, facing the sim-to-real gap. Alternatively, self-supervised methods rely on task-specific heuristics such as smoothness losses and cycle consistency [21, 4, 35], i.e., tracking a point forward in time, then backward, should return to its original position. In contrast, we use a zero-shot method to extract flow from general-purpose self-supervised video models trained on diverse, large-scale datasets, as they better learn challenging real-world dynamics (Section 6). Further, the best performing trackers on long horizon benchmarks often exploit multi-frame context, which can obscure a model's ability to resolve *core* frame-to-frame dynamics across variable timesteps. Therefore, we compare our zero-shot method primarily against baselines following the traditional, two-frame setting for flow.

**Deterministic Video Models** predict a *single* future frame/latent [15, 14, 16, 3, 2]. Trained with either latent or pixel $\ell_1/\ell_2$ reconstruction losses, they are implicitly optimized to output the *expectation* over plausible futures. Whenever the conditioning signal—a single frame $F_1$, multiple frames $(F_{-N} \ldots F_1)$, or a partially masked target $F_2^{\text{masked}}$—can lead to multiple future possibilities, the model averages them, producing spatially blurred predictions which hurt flow estimation.

**Generative Video Models.** Generative models can sample different image or video generations from a probabilistic distribution. Diffusion models iteratively denoise a single noise sample into a generation [19, 36, 33, 20, 5]. Autoregressive models [31, 7, 32, 8] generate outputs sequentially, modeling the joint distribution over pixels or tokens, each element conditioned on previously generated elements. However, many of these models, e.g., Cosmos [30], predict tokens in raster order and use a global tokenizer where each image patch is encoded with information from other patches. Hence, they lack fine-grained controllability which makes them challenging to use for flow extraction (Section 4.3) In contrast, a recent class of models, Local Random Access Sequence (LRAS), has several key properties which make them suitable for optical flow extraction (Section 4.4). Applying our novel test-time inference procedure to LRAS achieves state-of-the-art results (Section 6).

**Tracks from Diffusion Models.** Related to our work, there has been recent interest in using the emergent motion understanding capabilities of generative video models. The focus of these

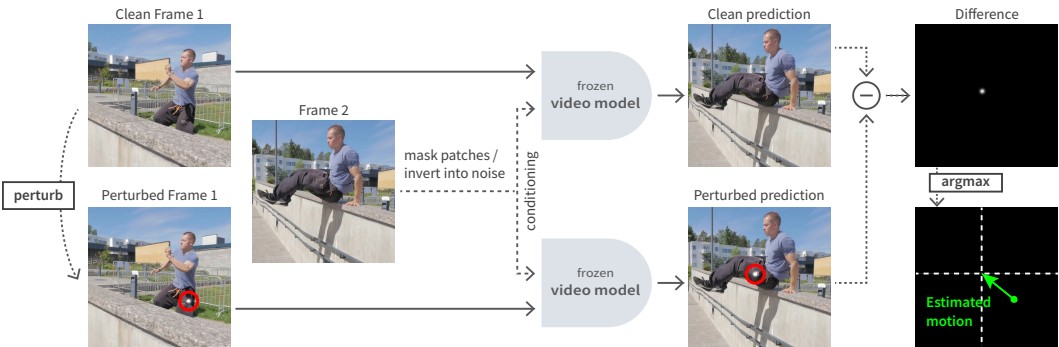

Figure 2: **Test-time inference procedure for extracting flow from a pre-trained, frozen, generative video model, based on the Counterfactual World Model (CWM) paper [3].** This involves three steps: (1) Perturbation: add a small, white-colored 2D Gaussian dot perturbation to frame 1 at the location of the point we wish to track. (2) Generate model predictions conditioned on the two frames. For CWM, Cosmos, and LRAS, we provide frame 1 and masked patches of frame 2 (Sections 4.1, 4.3, 4.4). For Stable Video Diffusion, we provide the noised latents of both frames (Section 4.2). (3) Estimate optical flow by computing the RGB difference between the clean and perturbed predictions.

investigations has primarily been on reading out motion information from intermediate representations of diffusion models [38, 29]. In this work, we instead rely on a different extraction procedure (namely, propagating intervention points for tracking), which we find to be more transferable to generative architectures beyond diffusion. Our approach, when paired with design decisions that maximize controllability and locality, outperforms the latest diffusion-specific methods for flow (Section 6).

# 3 Test-time inference procedure and evaluation setup for optical flow

We use a zero-shot procedure to extract optical flow from various pre-trained video models. For all models, we utilize the generic method template illustrated in Figure 2, based on the Counterfactual World Model (CWM) paper [3]. **Step 1. Inject a small perturbation.** We duplicate the initial frame $F_1$ and perturb it with a small "white bump" to form $\tilde{F}_1$, i.e., a Gaussian centered at the query location $x_q$ with amplitudes 255 on each RGB channel and standard deviation $\sigma$ equal to 2.0. **Step 2. Run model twice.** Both clean and perturbed initial frames are separately forwarded through the frozen model, each time with a sparse mask of the second frame ($F_2^{\text{masked}}$). Hence, two forward passes: $(F_1, F_2^{\text{masked}} \rightarrow F_2^{\text{pred}})$ and $(\tilde{F}_1, F_2^{\text{masked}} \rightarrow \tilde{F}_2^{\text{pred}})$. **Step 3. Estimate optical flow.** Compute the RGB difference between the two predictions, $F_2^{\text{pred}}$ and $\tilde{F}_2^{\text{pred}}$. Then, take the arg-max to identify the location to which the perturbation was carried.

We find that this procedure can be extended to any video model that exposes this interface of next-frame prediction. For example, this works with models that allow providing a subset or masking of patches (Sections 4.1, 4.3, 4.4), and also video diffusion models, by converting frames 1 and 2 into noised latents and using them as conditioning (Section 4.2).

**Evaluation setup.** We use TAP-Vid DAVIS [9] and Kubric [13] for evaluation. TAP-Vid DAVIS contains real-world videos with human-annotated flow and occlusion labels, while Kubric is a synthetic dataset with ground-truth labels. **Metrics.** We use the following metrics. (1) *Average Distance (AD)* (or endpoint error (EPE)), the Euclidean distance between the predicted and ground-truth flow, as well as metrics from TAP-Vid [9]: (2) *Average Jaccard* (AJ), the "true positives" divided by "true positives" plus "false positive" plus "false negatives", averaged over various thresholds; (3) $< \delta_{\text{avg}}^x$ the fraction of points that are within a threshold of the ground truth location, averaged over various thresholds, and (4) *Occlusion Accuracy (OA)* for predicting occluded points.

# 4 Model evaluations

We conduct several studies extracting optical flow from various video model classes, highlighting issues we found with each. We investigate deterministic models (Section 4.1), diffusion models

(Section 4.2), and autoregressive models (Section 4.3). Finally, we identify the key properties of generative models that aid precise flow prediction, and a model class that satisfies them (Section 4.4).

## 4.1 Study 1: Deterministic models produce blurry-averaged predictions.

**Method:** Deterministic video models such as the Counterfactual World Model (CWM) [3, 40] predict a single future state. CWM is a visual foundation model that can be zero-shot prompted to perform many tasks, such as predicting flow, keypoints, object segments, counterfactuals, and depth. CWM was trained with an $\ell_2$ loss to perform masked next-frame prediction: $(F_1, F_2^{\mathrm{masked}} \to F_2^{\mathrm{pred}})$. It deterministically predicts a single future state $F_2^{\mathrm{pred}}$, as it lacks a mechanism for sampling. To extract flow from CWM, we use the procedure in Section 3 with a small, red perturbation, following [3].

**Findings: Deterministic video models such as CWM regress to the mean future state, blurring predictions.** Deterministic models, such as the Counterfactual World Model (CWM), often produce blurry predictions as they model a single, average future state (Figure 4A). This hurts optical flow extraction in two ways. (1) At the location where the perturbation is carried to, the perturbation is less visible due to blurriness from prediction uncertainty. (2) At locations where the perturbation is *not* expected to propagate, blurriness manifests as minor RGB differences. The clean and perturbed predictions are different in these locations due to the added perturbation in the first frame ($F_1$). Combined, these introduce errors when using arg-max to select the destination for the query point. To address the issue of blurry predictions from CWM, we next explore extracting flow from Stable Video Diffusion (SVD), a diffusion model that produces sharp predictions.

## 4.2 Study 2: Diffusion models' global latent code lacks sufficient fine-grained controllability

**Method:** Applying the method above (Section 3, Figure 2) to Stable Video Diffusion [5] is challenging as generation operates on full-frame latents rather than pixels, so these models do not natively support partial frames as input. However, we need to provide information from the second frame to anchor the generation, otherwise the model will hallucinate arbitrary future frames unsuitable for flow estimation. Below, we describe how we adapt the flow extraction procedure to the latent diffusion setting, while keeping the generations anchored to the observed video. **Step 1. Frame selection and perturbation.** As SVD is trained with a fixed framerate and more than two frames, we first retrieve an interpolated sequence of intermediate frames bridging the query ($F_1$) and target ($F_N$) frames $\mathcal{F} = \{F_1, F_2, \ldots, F_N\}$. We add a dot perturbation to $F_1$, obtaining $\tilde{F}_1$. **Step 2. Latent inversion and paired generation.** To anchor the sampling trajectory to the original video, we apply a latent inversion technique introduced in VideoShop [10]. Inversion addresses the drift in generation by finding an initial noise vector that reconstructs the input video when fed into the diffusion process. Specifically, we first obtain the latents for every frame $\boldsymbol{\ell} = \{\ell_1, \ell_2, \ldots, \ell_N\}$ and then partially apply the inversion process, adding noise to the latents up to a fraction of the total noising steps. This is done using SVD's UNet and an inverted scheduler, resulting in noisy latents $\tilde{\boldsymbol{\ell}} = \{\tilde{\ell}_1, \ldots, \tilde{\ell}_N\}$. From these latents we perform two denoising passes with identical hyperparameters, but with different conditioning frames, $F_1$ and the perturbed $\tilde{F}_1$, obtaining two denoised latents $\boldsymbol{\ell}^{\mathrm{pred}}$ and $\tilde{\boldsymbol{\ell}}^{\mathrm{pred}}$, which can be decoded to produce the clean $\mathcal{F}^{\mathrm{pred}}$ and perturbed reconstruction $\tilde{\mathcal{F}}^{\mathrm{pred}}$. **Step 3. Estimate optical flow** (same as in Section 3). We compute the RGB difference between the clean and perturbed generations to localize the perturbation. We compare two generations, rather than the perturbed generation to the original input, in order to suppress noise from imperfect inversion and VAE artifacts, improving the accuracy and robustness of flow predictions even under lossy latent compression.

**Findings: Stable Video Diffusion (SVD) lacks fine-grained controllability as it generates each frame by globally denoising a coarse latent code.** Thus, local perturbations in pixel space are washed out or remapped unpredictably during sampling. This stochastic, scene-level regeneration prevents deterministic, point-wise correspondences. As a result, its clean and perturbed predictions differ in locations where the perturbation is *not* supposed to be carried to, resulting in noisy differences.

Attempting to address the precision issues arising from a coarse global latent, the recent DiffTrack [29] uses a one-to-one frame-to-latent mapping to avoid temporal compression. It uses query-key matching in the 3D attention blocks across frames to perform point tracking. Though this method demonstrates that strong temporal correspondence can be found in diffusion model representations, with significant improvement over existing diffusion-based methods such as DIFT [38], it still lags

behind specialized optical flow models, thus failing to close the gap between general-purpose video models and task-specific baselines (Table 3). Next, we explore autoregressive models which allow us to provide actual patches of the second frame as more fine-grained conditioning.

### 4.3 Study 3: Raster-order autoregressive models struggle with partial-frame conditioning

**Method:** Cosmos [30] includes both a diffusion-based and an autoregressive foundation model. We evaluate flow extractions from the autoregressive model which allows us to provide the actual frame patches directly as conditioning. We use the Cosmos autoregressive model with 4B parameters, which comes with a 7B diffusion decoder for generating images. As the Cosmos autoregressive model does not have pointer tokens, autoregressive rollout predictions for image tokens cannot be performed in random-access order. Instead, they can only be made in raster order, i.e., from top left to bottom right of the image. Nevertheless, we make a best-effort attempt to extract flow by trying multiple settings.

- *Setting A.* We provide the first frame and the top 10% raster tokens of the second frame: clean $(F_1, F_2^{\text{top-10\%-raster}} \rightarrow F_2^{\text{pred}})$ and perturbed $(\tilde{F}_1, F_2^{\text{top-10\%-raster}} \rightarrow \tilde{F}_2^{\text{pred}})$.
- *Setting B.* We provide the first frame and overwrite a random 10% of the predicted tokens during the model's autoregressive rollout with ground truth tokens of the second frame: clean $(F_1, F_2^{\text{overwrite-random-10\%}} \rightarrow F_2^{\text{pred}})$ and perturbed $(\tilde{F}_1, F_2^{\text{overwrite-random-10\%}} \rightarrow \tilde{F}_2^{\text{pred}})$.
- *Setting C.* We provide both frames fully: clean $(F_1, F_2 \rightarrow F_2^{\text{pred}})$ and perturbed $(\tilde{F}_1, F_2 \rightarrow \tilde{F}_2^{\text{pred}})$.

**Findings: All Cosmos evaluation settings perform poorly** (Figure 7 and Table 2).

- *Setting A.* The model prediction effectively repeats the first frame, unable to propagate the perturbation (Figure 7A). This is because the top 10% raster patches reveal less about the second frame than random 10% patches. At the same time, revealing more patches will hurt performance; if the perturbation should be carried to a revealed patch, the model will not generate the perturbation.
- *Setting B.* This also results in the model prediction repeating the first frame, because the model generates many new patches with less than 10% ground truth patches. This is much more challenging than getting all the revealed ground truth patches at the start.
- *Setting C.* The model prediction correctly matches the target frame but does not contain the perturbation. We evaluate this setting as the model has a diffusion decoder that has the potential to (but unfortunately does not) reproduce the perturbation in $\tilde{F}_2^{\text{pred}}$.

Cosmos flow extraction is challenging due to the lack of a non-local tokenizer and pointer tokens for random-access decoding order. Therefore, we next explore LRAS, which has these properties.

### 4.4 Study 4: Key model properties for flow extraction are present in LRAS

**Method:** From Sections 4.1, 4.2, and 4.3, we discover that flow extraction can be improved by generative models having three key properties (Table 1): (1) distributional prediction of future frames (avoiding blurry or noisy outputs; which CWM lacks); (2) local tokenizer that treats each spatio-temporal patch independently (which SVD and Cosmos lack); and (3) random-access decoding order that allows the model to condition on any subset of the second frame patches (which SVD and Cosmos lack). We identify Local Random Access Sequence (LRAS), a recent family of autoregressive visual foundation world models that has these properties and can be zero-shot prompted to perform many tasks, such as 3D object manipulation, novel view synthesis, and depth estimation. We use its autoregressive generative video world model with 7 billion (7B) parameters.

**Findings: LRAS model produces clean and perturbed predictions with minimal variations due to sampling randomness.** However, these small variations result in slightly noisy difference maps, harming flow extraction (Figure 5). We address this with the KL-tracing procedure below.

## 5 KL-tracing bypasses sampling randomness

**Method:** We design a novel test-time inference procedure, *KL-tracing*, for optical flow extraction from models that predict a probability distribution at every patch of the target frame (Figure 3). KL-tracing is the same as the RGB flow extraction method in Section 3, except for the third step. **Step 3. Estimate optical flow with patchwise KL-divergence.** We use the patch-wise logits for the clean $(F_2^{\text{pred}})$ and perturbed $(\tilde{F}_2^{\text{pred}})$ predictions: $\{\mathbf{z}_{ij}\}$ and $\{\tilde{\mathbf{z}}_{ij}^{\text{pert}}\}$ (Figure 3). For every patch

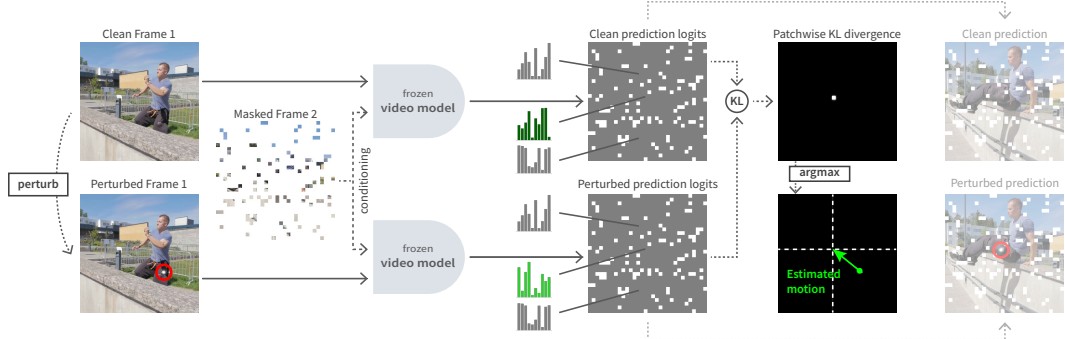

Figure 3: ***KL-tracing*, our novel yet simple test-time inference procedure for extracting optical flow from controllable generative models such as LRAS.** We follow the same steps for perturbation and conditioned prediction as in Figure 2, but estimate optical flow by computing the KL divergence between the clean and perturbed prediction logits.

$(i, j)$ we compute the KL-divergence: $D_{\mathrm{KL}}(i, j) = \mathrm{KL}\big[(\mathbf{z}_{ij}) \,\|\, (\tilde{\mathbf{z}}_{ij}^{\mathrm{pert}})\big]$. The resulting KL divergence map should peak at the patch where the perturbation is carried to, giving a crisp optical flow estimate (Figure 3). We take an arg-max to identify the patch with the highest KL divergence to estimate optical flow. To detect if a point is occluded, we simply use a threshold on the KL divergence value.

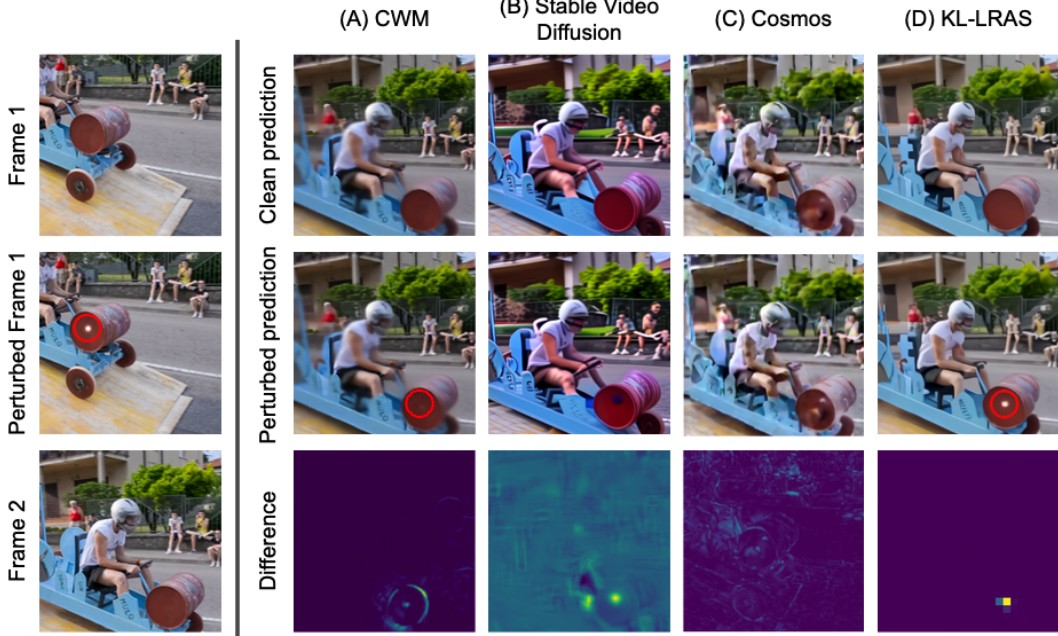

Figure 4: **Our method, KL-tracing using LRAS extracts better flow than other generative video models.** (A) Deterministic models, such as CWM [3], often produce blurry predictions as they model a single, average future state. (B) Stable Video Diffusion lacks fine-grained controllability due to its coarse global latent code. Its clean and perturbed predictions differ in locations where the perturbation is *not* supposed to be carried to. (C) The Cosmos autoregressive world model lacks fine-grained controllability as it does not utilize pointers to denote the position of each token, making it challenging to prompt for flow extraction. (D) The LRAS model is highly controllable and has minimal differences between the clean and perturbed predictions. We use KL-tracing to compute the difference in logit instead of RGB space, obtaining sharp flow extractions.

**Findings: KL divergence of prediction logits bypasses noisy differences resulting from sampling randomness.** Each RGB generation is a sample drawn from the prediction distribution. Instead of sampling multiple RGB generations and averaging them to suppress the noisy differences resulting

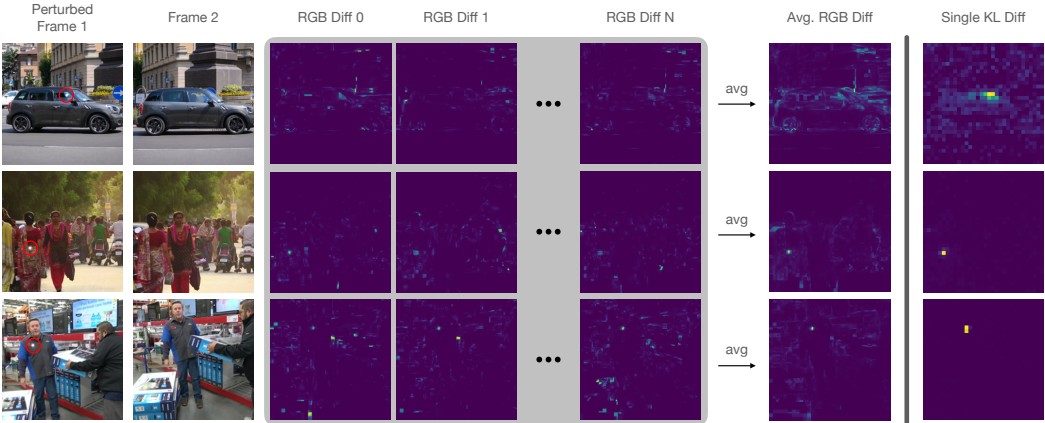

Figure 5: **KL-divergence of prediction distributions bypasses noisy RGB differences resulting from sampling randomness.** Computing the KL divergence of the clean and perturbed prediction logits (last column) is more efficient yet functionally similar to computing the average RGB difference over many samples (second last column).

from sampling randomness, it is more efficient to directly use the predicted distribution. This is an important benefit of autoregressive models as they directly expose the probability distribution. Empirically, we observe two benefits. First, there are minor differences between the clean and perturbed predictions at locations where the perturbation is *not* expected to propagate, because adding the perturbation affects the rest of the image due to the transformer's global attention. KL-tracing empirically results in smaller noisy differences than computing the RGB difference. Second, occasionally the added perturbation does not visually get carried over in the perturbed RGB prediction, but it still appears as elevated *uncertainty* in logit space, detected by KL-tracing.

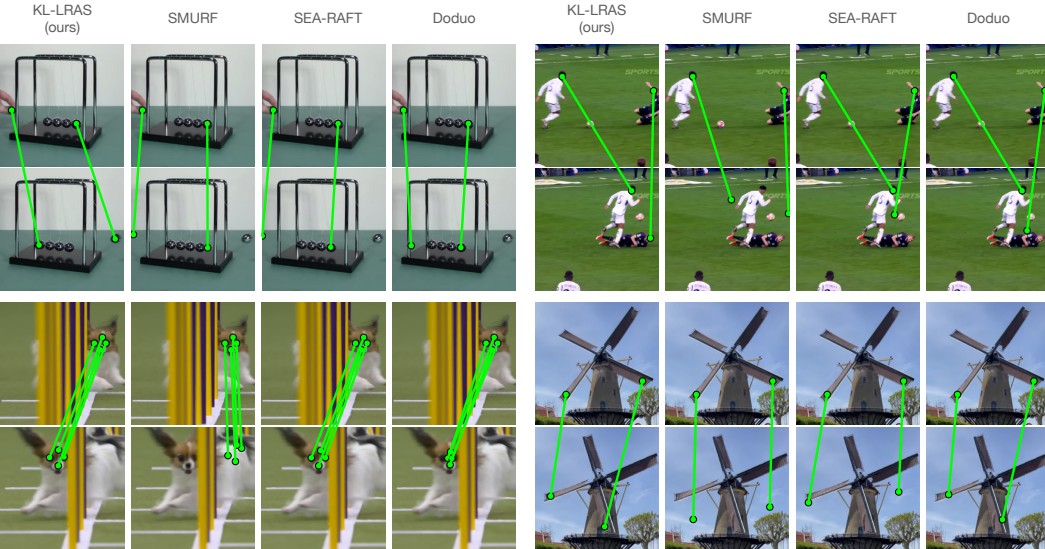

Figure 6: **Large world models such as LRAS capture aspects of real-world dynamics that are challenging for specialized flow models relying on visual similarity or photometric loss.** Specialized optical flow baselines, both supervised and self-supervised, struggle on complex, real-world scenes that are not fully captured by their training heuristics (Section 6). A deeper understanding of dynamics beyond feature/photometric similarity allows large world models such as LRAS to resolve motion in the presence of homogeneous objects, motion blur, and partial occlusion, among others.

# 6 Large video world models capture aspects of real-world dynamics that are challenging for specialized flow models

**Method:** We compare KL-tracing with supervised flow methods such as RAFT [39] and SEA-RAFT [41], as well as unsupervised methods such as Doduo [22] and SMURF [37] (Table 3, Figure 6). We also compare our method to an alternative branch of approaches, multi-frame trackers, which typically use multiple frames of information to generate point tracks across time (Appendix C).

**Findings: Powerful video models capture challenging aspects of real-world dynamics.** For example, a globe undergoes in-place rotation with the query point on a textureless surface (Figure 1), or physical motion examples such as Newton's cradle (Figure 6). These real-world dynamics are challenging for specialized flow models which are small and not trained on diverse, large-scale video datasets. **Video world models use generic training objectives, so they do not suffer from failure modes introduced by training heuristics used by specialized flow models.** Supervised methods are trained on synthetic datasets as labeling real-world videos is expensive, and hence face a sim-to-real gap where they do not observe challenging real-world dynamics. Self-supervised methods such as Doduo [22] and SMURF [37] use photometric loss for training, which results in challenges in predicting flow for frame pairs with significant differences in light intensity, and also enforce strong global consistency and smoothness constraints, which introduce failure modes for complex non-uniform dynamics that contain various magnitudes of flow across the same frame pair.

|                                          | CWM [3] | SVD [5] | Cosmos [30] | LRAS [25] |
|------------------------------------------|---------|---------|-------------|-----------|
| Distributional prediction of future frames | No      | **Yes** | **Yes**     | **Yes**   |
| Non-global tokenizer                     | **Yes** | No      | No          | **Yes**   |
| Random-access decoding order             | **Yes** | No      | No          | **Yes**   |

Table 1: **Key properties of video models for precise flow extraction**: (1) distributional prediction of future frames, thus avoiding blurry or noisy outputs, (2) non-global tokenizer that treats each spatio-temporal patch independently, (3) random-access decoding order that allows conditioning on any subset of second-frame patches.

| Model | TAP-Vid DAVIS Subset (3%) Endpoint Error (EPE) |
|-------|------------------------------------------------|
| LRAS RGB (5MM, 8MS, 2STD) (ours) | 8.4797 |
| LRAS KL (5MM, 8MS, 2STD) (ours) | **5.0762** |
| Stable Video Diffusion [5] | 74.7990 |
| Cosmos (top 10% raster) (5MM, 2STD, 512×512) [30] | 35.4338 |
| Cosmos (overwrite 10% during rollout) (5MM, 2STD, 512×512) [30] | 37.7552 |
| Cosmos (provide full second frame) (5MM, 2STD, 512×512) [30] | 66.5521 |

Table 2: **KL-tracing** with LRAS beats other video models. MM = multi-mask (average over multiple random masks), MS = multi-scale (zoom into image at multiple scales).

# 7 Discussion

**Taming generative world models with fine-grained controllability.** A key finding of our work is that zero-shot prompting many mainstream generative video models for visual property extraction can be challenging due to their lack of fine-grained controllability. However, a series of design features can help tame these models: (1) predicting a distribution of future states rather than a deterministic average, (2) encoding each spatio-temporal patch independently, and (3) having a mechanism to decode individual parts of a frame in any given order. Currently, the LRAS [25] framework was the only one we identified that contained all of these properties, but they could be adopted by other generative models as well, increasing the ease with which they can be visually prompted.

**Tight coupling with the base generative model.** The main advantage of using a generative model for flow is that it becomes easier to improve flow prediction quality on new domains. Any previously

| | Method | DAVIS | | | | Kubric | | | |
|---|---|---|---|---|---|---|---|---|---|
| | | AJ ↑ | AD ↓ | $<\delta^x_{\text{avg}}$ ↑ | OA ↑ | AJ ↑ | AD ↓ | $<\delta^x_{\text{avg}}$ ↑ | OA ↑ |
| *TAP-Vid First* | | | | | | | | | |
| S | RAFT [39] | 41.77 | 25.33 | 54.37 | 66.40 | 71.93 | 5.60 | 82.15 | 88.54 |
| | SEA-RAFT [41] | 43.41 | 20.18 | 58.69 | 66.34 | 75.06 | 6.54 | 84.63 | 89.50 |
| W | Doduo [22] | 23.34 | 13.41 | 48.50 | 47.91 | 54.98 | 5.31 | 72.20 | 73.56 |
| U | SMURF [37] | 30.64 | 27.28 | 44.18 | 59.15 | **65.81** | 6.81 | 80.57 | **87.91** |
| | DiffTrack [29] | - | - | 46.90 | - | - | - | - | - |
| | CWM [3, 40] | 15.00 | 23.53 | 26.30 | **76.63** | 28.77 | 11.64 | 41.63 | 84.93 |
| | LRAS with KL-tracing (ours) | **44.16** | **11.18** | **65.20** | 74.58 | 65.49 | **5.06** | **81.66** | 87.63 |

Table 3: **TAP-Vid First: quantitative results on DAVIS and Kubric.** Tracking starts when a point first appears and continues to the video end, thus involving large frame gaps. LRAS with KL-tracing outperforms two-frame baselines. "S" = supervised, "W" = weakly supervised, "U" = unsupervised.

unrepresented domain can become "in-distribution" for KL-tracing, simply by extending the training data of the base generative model (e.g., LRAS). There is no need to obtain new, task-specific labels, which is particularly appealing for flow due to the intractable cost of human labeling for anything other than synthetic, computer-graphic type scenes. As modern video datasets continue to scale at a rapidly increasing rate, KL-tracing's performance can improve in lockstep.

**Distillation.** Compared with real-time dense flow baselines [39, 41] or diffusion-based methods that build on feature correspondence [38, 29], KL-tracing can be slower due to the process of propagating interventions point-by-point. However, on challenging, real-world scenes, the quality of the extracted flow is notably better (Table 3). In fact, we can improve the Pareto frontier by taking the sparse labels that are extracted and distilling them into a faster architecture such as SEA-RAFT, thereby using KL-tracing as a much more scalable alternative to human labeling.

**Future work.** A natural next step is extending this test-time inference procedure to extract other visual intermediates such as object segments, material properties, or motion affordances. By moving away from fine-tuning our world models on labeled datasets to a paradigm of zero-shot prompting, the vision community can experience a shift akin to the one seen with large language models, moving from rigid representations constructed by fine-tuning toward dynamic, task-specific and adaptable ones extracted on the fly.

Overall, our results indicate that prompting controllable, self-supervised world models is a scalable and effective alternative to supervised or photometric-loss approaches for high-quality optical flow.

## Acknowledgments and Disclosure of Funding

This work was supported by the following awards: Simons Foundation grant SFI-AN-NC-GB-Culmination-00002986-05, National Science Foundation CAREER grant 1844724, National Science Foundation Grant NCS-FR 2123963, National Science Foundation Grant RI 2211258, Office of Naval Research grant N00014-20-1-2589, ONR MURI N00014-21-1-2801, ONR MURI N00014-24-1-2748, and ONR MURI N00014-22-1-2740. We also thank the Stanford HAI, Stanford Data Sciences, the Marlowe team, and the Google TPU Research Cloud team for their computing support.

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

# A  Additional details on methods

| Models | Perturbation | Predictions based on conditioning | Compute difference |
|---|---|---|---|
| CWM [3] | White Gaussian | Frame 1, Partial Frame 2 | RGB difference |
| SVD [5] | White Gaussian | Noised latents of Frame 1 and 2 | RGB difference |
| Cosmos [30] | White Gaussian | Frame 1, Partial Frame 2 | RGB difference |
| LRAS, RGB [25] | White Gaussian | Frame 1, Partial Frame 2 | RGB difference |
| LRAS, KL | White Gaussian | Frame 1, Partial Frame 2 | KL divergence of logits |

Table 4: **Flow extraction for all models follows the same three-step template:** (1) Perturbation of Frame 1 with a small, white-colored Gaussian bump, (2) Conditioning the model on frames 1 and 2 to generate clean and perturbed predictions, and (3) Compute the difference between the clean and perturbed predictions to extract optical flow.

# B  Additional results

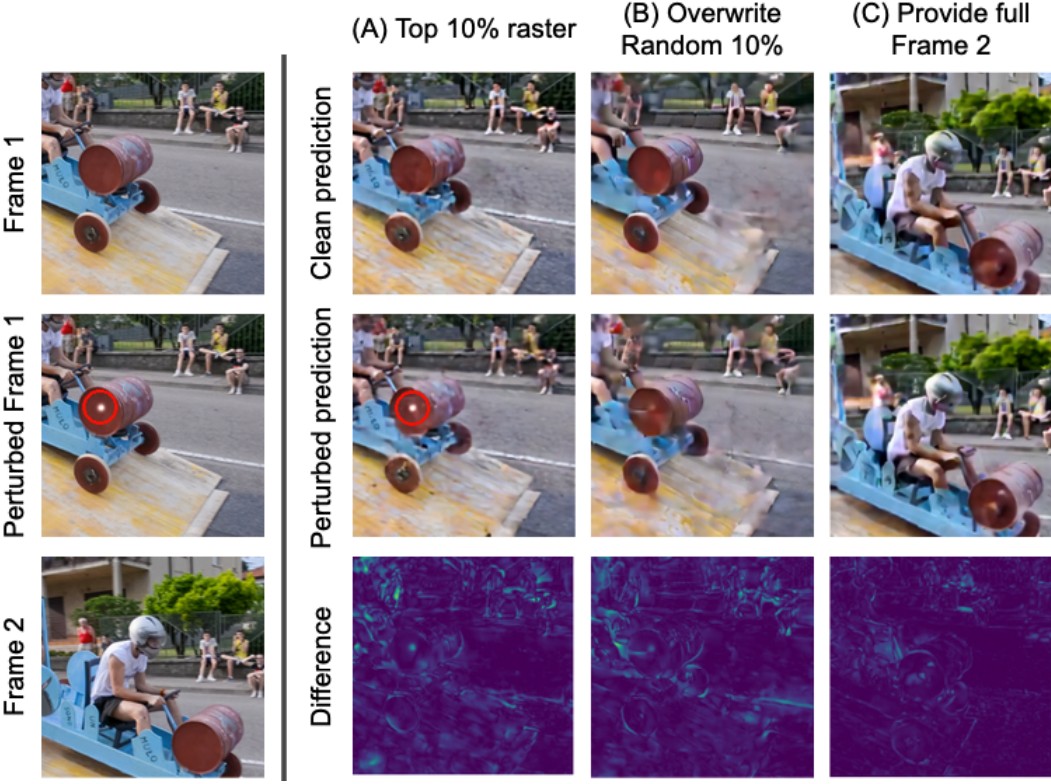

Figure 7: **All evaluation settings for Cosmos [30]) result in poor flow extractions.** See Section 4.3 for a more detailed explanation of each result.

# C  Multi-frame trackers

We are primarily interested in evaluating a model's ability to resolve motion from two frames, with methods and baselines chosen accordingly. While multi-frame trackers often achieve improved results due to aggregating and refining predictions globally, the two-frame setting directly evaluates a model's core dynamics understanding, especially in more challenging real-world cases. We compare our method with supervised, multi-frame trackers run under the same two-frame constraint.

| Method | DAVIS | | |
|---|---|---|---|
| | AJ $\uparrow$ | $< \delta^x_{\text{avg}} \uparrow$ | OA $\uparrow$ |
| *TAP-Vid First* | | | |
| S  CoTracker-v3 [24] | 39.9 | 58.0 | 76.9 |
|    AllTracker [18] | **54.9** | **66.0** | **78.0** |
| U  LRAS with KL-tracing (ours) | 44.2 | 65.2 | 74.6 |

Table 5: **TAP-Vid DAVIS results for two-frame trackers.** Tracking starts when a point first appears and continues to the video end (large frame gaps). The supervised tracker baselines are natively multi-frame, but are evaluated in a two-frame setting for direct comparison.

The performance of multi-frame trackers is significantly impacted when removing context from intermediate frames. Notably, we find that AllTracker [18] is more robust, due to its approach of merging a two-frame, flow-type correspondence with a tracker-style global refinement across frames. Despite being completely training-free, our method shows comparable performance with the latest trackers in this challenging setting (Table 5).

