# OpenReview forum: "Taming generative video models for zero-shot optical flow extraction"
_NeurIPS.cc/2025/Conference — NeurIPS 2025 poster_

### Official Review · Reviewer_nvPE · 2025-06-28

**Clarity:** 3
**Significance:** 3
**Originality:** 3
**Rating:** 5
**Confidence:** 5

**Summary:**

This paper introduces KL-tracing, a new approach to tracking image-space perturbations, for unsupervised optical flow extraction from the LRAS model. This paper also designs several new baselines from existing video models, such as stable video diffusion and cosmos. KL-tracking-based LRAS achieves strong performance on TAP-Vid, outperforming existing unsupervised models and some flow-specific models.

**Questions:**

See above.

**Ethical Concerns:**

["NO or VERY MINOR ethics concerns only"]

**Final Justification:**

My concerns are addressed. I raise my rating to "Accept".

**Limitations:**

Yes

**Quality:**

3

**Strengths And Weaknesses:**

### Strengths
- From reported results, KL-tracing is more effective than RGB-difference-based perturbation tracking. It significantly improves the performance.
- Authors also design baselines from other well-known video generative models, making the comparisons more fair.

### Minor Weaknesses
- The baselines from other models (e.g. cosmos) are relatively weak.
- The overall design only allows 2-view point-tracking in a single inference, while optical flow is usually evaluated on (semi-)dense pixels. I recommend authors to follow the standard optical flow settings and evaluate KL-tracing on Sintel[1], KITTI[2], and Spring[3].

[1] Butler, Daniel J., et al. "A naturalistic open source movie for optical flow evaluation." Computer Vision–ECCV 2012: 12th European Conference on Computer Vision, Florence, Italy, October 7-13, 2012, Proceedings, Part VI 12. Springer Berlin Heidelberg, 2012.

[2] Geiger, Andreas, Philip Lenz, and Raquel Urtasun. "Are we ready for autonomous driving? the kitti vision benchmark suite." 2012 IEEE conference on computer vision and pattern recognition. IEEE, 2012.

[3] Mehl, Lukas, et al. "Spring: A high-resolution high-detail dataset and benchmark for scene flow, optical flow and stereo." Proceedings of the IEEE/CVF Conference on Computer Vision and Pattern Recognition. 2023.

---

> ### Author Rebuttal · Authors · 2025-07-31
>
> We thank the reviewer for the thoughtful review and constructive suggestions! Overall, we are glad that reviewers found our method "novel" and "interesting" (uSVM) and our empirical results and analyses to be "strong" and "detailed," and a significant improvement to existing methods (fTmd, uSVM, nvPE). We address the main points of the reviewer below:
>
> > The baselines from other models (e.g., cosmos) are relatively weak.
>
> Flow improvement is primarily measured by comparing to stronger, task-specific baselines such as SEA-RAFT (Table 2), against which our method achieves better performance. The results from Table 1 that showcase Cosmos and SVD separately demonstrate the challenge of extracting motion cues from large-scale generative models that lack controllability. We will integrate Cosmos and SVD results into the main TAP-Vid table (Table 2) in future iterations for a more comprehensive comparison. Further, we introduce a significantly stronger (though more architecture-specific) baseline, DiffTrack [1], that also utilizes Diffusion models, and note that our method still achieves higher performance.
>
> | TAP‑Vid DAVIS                 | $<\delta^x_{\text{avg}}\uparrow$   |
> |:-----------------------------|-----------------:|
> | SEA‑RAFT                     | 58.7 |
> | DiffTrack [1] |        46.9 |
> | KL‑LRAS (ours)               |     **65.2** |
>
> > The overall design only allows 2-view point-tracking in a single inference, while optical flow is usually evaluated on (semi-)dense pixels. I recommend authors to follow the standard optical flow settings and evaluate KL-tracing on Sintel, KITTI, and Spring.
>
> Thank you for this important suggestion. As our main focus is a generalizable flow extraction method for real-world videos, we chose TAP-Vid as our main benchmark, following a recent trend [2, 3], as we believe it better captures a model's ability to generalize to challenging, in-the-wild scenes. To further improve our coverage, we also include results on Kubric (Table 2) from the same benchmark, which is a synthetic dataset similar to Sintel or Spring. We appreciate the suggestion for additional benchmarks and will consider expanding our evaluation in future work.
>
> [1] Nam et al., 2025. "Emergent Temporal Correspondences from Video Diffusion Transformers."
>
> [2] Jiang et al., 2023. "Doduo: Learning Dense Visual Correspondence from Unsupervised Semantic-Aware Flow."
>
> [3] Karaev et al., 2024. "CoTracker3: Simpler and Better Point Tracking by Pseudo-Labelling Real Videos."

---

> > ### Comment · Reviewer_nvPE · 2025-08-05
> >
> > Thank the reviewer for detailed responses. Most of my concerns are addressed. I also checked the performance on TAP-Vid-DAVIS compared to 2-view CoTrackerv3. However, the 2-view setting significantly weakens the CoTrackerv3. A comparison between KL-LARS and AllTracker[1] will be more persuasive.
> >
> > I keep the original score now and will raise it if this concern is addressed.
> >
> > [1] Harley, Adam W., et al. "AllTracker: Efficient Dense Point Tracking at High Resolution.", ICCV 2025

---

> ### Author Response · Authors · 2025-08-05
> **trying to understand better**
>
> Wait, something is a bit confusing here for us.  KL tracing (and the other methods we investigate in our paper) are all *optical flow* methods, not tracking methods.  That means, by definition, they apply to the 2-frame case.  We set out to solve the problem of optical flow, not multi-frame tracking, to begin with.   Doesn't that mean the usual 2-frame case is the proper evaluation for our question setting?   (Maybe it was confusing when we used the term "tracing" with KL-tracing?  It wasn't meant to imply that we were attempting to build a tracking solution, but maybe the words 'tracing' and "tracking" sound similar and that was confusing?)
>
> One good way to think about this is:  the 2-frame setting is the "core dynamics understanding" problem.  It's harder than the multi-frame case.  And thus all methods, whether they are co-tracker or KL-tracing or RAFT or whatever, will naturally post lower absolute numbers in the 2-frame setting as compared to the multi-frame setting, where it's possible to average over information from the multiple frames to boost the numbers.    The 2-frame setting isn't "unfair", it's just different, and equally important, to the multi-frame setting.
>
> We've shown, with our 2-frame co-tracker comparison, that the supervised co-tracker method doesn't have some magical ingredient that makes it better at the base, hard optical flow case.   (That's what we though you would care about, since our whole paper is in the optical flow domain rather than tracking, to begin with.) In other words, we've shown that whatever higher numbers Cotracker is getting in the multi-frame case comes from the additional information of the multiple frames, not some deeply better dynamics-understanding principle.  That's a fair comparison for us to make, and allows us to put that method on the same footing as all the other optical flow methods that we investigate in the paper (not just KL-tracing).
>
> Are you are kind of saying we should have been trying to solve multi-frame tracking the whole time, and that you think the 2-frame case is somehow "less valid" and really shouldn't have been what we were trying to go for in the first place?  Like are you saying that the multi-frame tracking case is somehow the "real standard" that we all should be testing against?
>
> Or is your point that we should be evaluating a multi-frame version of the KL-tracing method -- and all the other methods we investigate -- against multi-frame trackers?   Would that mean you think we should as part of this current paper have developed a multi-frame version of all the various methods we were testing in our paper?  That's actually a pretty good idea -- it probably should be done in the medium term, because typically a good 2-frame optical method can usually be turned into a good multi-frame tracker with some additional work -- but that seems to us to be a somewhat different research project than the one we presented (one focused on tracking rather than optical flow), and not the sort of thing one can do in the time frame of a rebuttal.
>
> One question about All-Tracker -- it looks like a cool method, but isn't it a supervised architecture that is pretty task-specialized for multi-frame tracking? It seems like it would be pretty far afield to compare self-supervised zero-shot optical flow methods to supervised multi-frame tracking methods.   I mean, naturally a supervised multi-frame model will be "better" in some absolute sense than a 2-frame zero-shot method, but is that comparison very meaningful?

---

> > ### Author Response · Authors · 2025-08-05
> > **making post visible to reviewer nvPE.**
> >
> > Sorry, reviewer nvPE! Just wanted to make sure you saw the above comment, since by accident the first time we posted it, we might not have made is visible to you.

---

> > ### Comment · Reviewer_nvPE · 2025-08-06
> >
> > No worries at all! Let's discuss. The paper is interesting and I'm positive. However, I think a higher score needs more careful study.
> >
> > My concerns come from: TAP-Vid-DAVIS is a standard dataset for point tracking instead of optical flow. I think it is non-trivial to upgrade KL-tracking to (semi-)dense cases for now, so the comparisons to current optical flow methods look weak to me.
> >
> > The reason why I ask about AllTracker is that it is more related to optical flow than CoTracker. Since the authors provide 2-view CoTracker results, I wonder if the authors can also provide 2-view AllTracker results.
> >
> > If my proposal above requires too much efforts, I wonder if authors can evaluate KL-tracing on sparse semantic correspondences[1] (e.g. SPair-71k).
> >
> > [1] Zhang, Junyi, et al. "A tale of two features: Stable diffusion complements dino for zero-shot semantic correspondence.", NeurIPS 2023

---

> > > ### Author Response · Authors · 2025-08-06
> > > **Replying to reviewer nvPE + AllTracker Results**
> > >
> > > We thank the reviewer for the thoughtful discussion!  We wanted to address two things:
> > >  1. **The issue of dense prediction.** The reviewer is right that having a fast dense predictor requires additional work relative to what we presented in the submission. It turns out that solving this using distillation has been pretty straightforward. We posted a discussion about this in response to one of the other reviewer's comments (XcXT), but we summarize it below for convenience:
> > >
> > >     > "As discussed in our section on limitations, the natural next step to take after successful flow extraction—as is common in the literature on using general foundation models to build deployable, task-specific models—is to distill it into a faster architecture that can output dense flow in real time. The main purpose of our method therefore is to be used as a sparse pseudo‑labeler. This will significantly increase domain diversity for flow datasets, which have traditionally been limited to synthetic, or heavily curated scenes, without incurring the intractable cost of human labeling.
> > >
> > >     >Since submission, we have been actively exploring distillation strategies, where we too have been interested in understanding the effect of label sparsity. In particular, we have trained an existing, off‑the‑shelf dense flow model on a synthetic dataset with ground‑truth labels using different levels of sparsity. We find that a model trained on an extremely sparsified (0.03% of points labeled) dataset performs competitively to the same model trained on the original, dense dataset (endpoint error: 2.47 for dense, 2.63 for sparse), strongly encouraging us on the likely effectiveness of sparse distillation.  Thus, it seems like distillation will end up being super effective and efficient, since almost perfect reconstruction of dense labels can be generated with very very sparse labels.  Our goal is to have the distilled model ready for release with the NeurIPS camera-ready version, if accepted.
> > >
> > >     >That distillation work very well is maybe not super super surprising to us since it has been observed for a while, kind of internally in the flow community, that fast feedforward flow extractors can be trained pretty sparsely -- and this fact is one of the reasons we've been ok with taking the zero-shot generative model approach."
> > >
> > > 2. **AllTacker results.**  We were able to download and run Alltracker (the code was easy to get, and no difficulties were encountered in running it), in the same two-frame setting as all the other methods. See results table below. For a more insightful comparison, we break down each result by frame gap (i.e. <= N indicates that only points with frame gap within N were evaluated), with "Full" being the full TAP-Vid First benchmark.
> > >   | TAP‑Vid DAVIS | AJ $\uparrow$ | $<\delta^x_{\text{avg}}\uparrow$ | OA $\uparrow$ |
> > >   |:--------------|----:|---------------------------:|----:|
> > >   | $\leq1$ | | | |
> > >   | AllTracker | 67.6 | 72.8 | 99.2 |
> > >   | KL-LRAS | 88.9 | 94.1 | 97.7  |
> > >   | $\leq2$ | | | |
> > >   | AllTracker | 58.8 | 65.9 | 98.9 |
> > >   | KL-LRAS | 86.2 | 92.6 | 97.0 |
> > >   | $\leq4$ | | | |
> > >   | AllTracker | 48.3 | 56.7 | 96.8 |
> > >   | KL-LRAS | 81.1 | 89.4 | 95.1 |
> > >   | $\leq8$ | | | |
> > >   | AllTracker | 35.9 | 45.0 | 92.5 |
> > >   | KL-LRAS | 74.2 | 85.3 | 92.0 |
> > >   | $\leq16$ | | | |
> > >   | AllTracker | 25.6 | 34.3 | 88.1 |
> > >   | KL-LRAS | 64.9 | 79.2 | 87.5 |
> > >   | $\leq32$ | | | |
> > >   | AllTracker | 18.4 | 26.7 | 82.1 |
> > >   | KL-LRAS | 53.6 | 71.8 | 80.5 |
> > >   | **Full** | | | |
> > >   | AllTracker | 13.1 | 20.3 | 75.8 |
> > >   | KL-LRAS | 44.2 | 65.2 | 74.6 |
> > >
> > >     The AllTracker results below reveal a similar story to CoTracker: namely, multi-frame trackers tend to leverage information from the entire video, usually resulting in significantly improved numbers on these benchmarks compared to the 2-frame case. Even at short frame gaps (e.g., 1,2,4), the default AllTracker uses information from future frames of the video to perform tracking. This is why when run under a 2-frame constraint (table above), the performance degrades significantly for AllTracker. Again, not to say that one setting is "fairer" than the other, but just that they emphasize different things.
> > >
> > >     The 2-frame constraint under which we evaluated all our baseline models directly evaluates a model's ability to understand the "core case" of challenging, real-world dynamics.  2-frame optical flow is of course useful for a bunch of applications, and performance numbers like these speak to the utility of algorithms (shch as KL-LRAS) in those applications.  There is also a decent chance that 2-frame improvements here can transfer to the multi-frame domain and result in an equally improved tracker, though showing that will have to be a subject for future work...
> > >
> > > We really appreciate the reviewer taking the time to go through this discussion with us in such careful detail and for being so engaged!

---

> > > > ### Comment · Reviewer_nvPE · 2025-08-06
> > > >
> > > > Thank you for the detailed responses! The comparisons to AllTracker look persuasive to me. I will raise my rating.

---

### Official Review · Reviewer_XcXT · 2025-06-30

**Clarity:** 2
**Significance:** 1
**Originality:** 1
**Rating:** 3
**Confidence:** 4

**Summary:**

The paper proposes a test-time intervention-based method to obtain the motion of one point for a future timestep in a video using an off-the-shelf video world model. The method works by introducing an intervention at the source point and then tracking that intervention's impact to subsequent frames. Two variants are explored, one where the effect is measured from changes in the output frames directly, and one where the effect is measured by the effect on the predicted posterior distributions.

**Questions:**

Currently, I see little evidence for the claims made for the main contributions 1 and 3, and in l. 52..54: Regarding main contribution 1, in the weaknesses section, I provided a non-exhaustive list of a range of methods that already investigate optical flow/tracker extraction from large generative (video) models, raising questions about the claim of this being "the first systematic study" of this topic. Given that the studies referenced also find other powerful mechanisms for optical tracking in such models, l. 52..54 seem unjustified given the analyses performed in this work. Similarly, just not comparing to relevant prior art does not make the proposed method state-of-the-art as claimed in main contribution 3.

Bringing this work to an acceptable level would require major revisions that thoroughly address the mentioned weaknesses.

**Ethical Concerns:**

["NO or VERY MINOR ethics concerns only"]

**Final Justification:**

Overall, I continue to lean towards rejection after considering the rebuttal, the other reviews, and the discussion. *Iff* the improvements promised by the authors, I am not fundamentally opposed to the acceptance of the paper. One major concern is that many of my concerns relate to a major misrepresentation of the contribution and main insights of the paper in the initial submission, whose resolution cannot be verified during the rebuttal/discussion phase.

Here is a list of the initially mentioned weaknesses I consider resolved (Y), partially resolved (~) or unresolved (X):
- W1 (major, X): The authors provided a speed comparison during the rebuttal, which shows the method to be 6+ orders of magnitude slower than sota methods. I consider this a severe limitation for any practical application
- W2 (major, Y): Mostly resolved
- W3 (major, ~): Resolving this concern can only be done by a fundamental revision of the presentation of the results and claims made in the paper, which can not be checked during the discussion period.
- W4 (minor, ~): Again primarily about presentation, so not verifiable
- W5 (minor, ~): I, unlike the authors, consider applicability to readily available models an important factor. Arguments can be made in both directions
- W6 (minor, ~): Again primarily about presentation, so not verifiable

**Limitations:**

No. The checklist refers to Sec. 5 for limitations, but I see *no* discussion of limitations of the proposed method in that section.

**Paper Formatting Concerns:**

The submitted PDF contains both content after the 9th page [p. 12] and a separate appendix in the supplementary material.

**Quality:**

1

**Strengths And Weaknesses:**

**Strengths**

The proposed method is simple, which makes it likely to generalize well to other models that fit the scope of base models covered by the method.



**Weaknesses**

1. Speed: the method only queries a single point per forward pass, which will likely make anything but the sparsest inference very costly. At the very least, an analysis of the quality/speed tradeoff compared to other methods should be provided.
1. Insufficient comparisons: the proposed method is not the first to enable (training-free) flow/tracker estimation based on pretrained generative vision models, so comparisons to these works should be included. Specifically, Track4Gen [Jeong et al., CVPR 2025] already evaluates a wide range of settings including zero-shot-, test-time optimization-, and full finetuning-based versions. Similarly, methods like DIFT [Tang et al., NeurIPS 2023] have also already demonstrated strong tracking performance using just features from image diffusion models. There are also a range of other works that investigated such mechanisms, such as Emerging Tracking from Video Diffusion [Zhang et al., OpenReview 2024], MOFT [Xiao et al., NeurIPS 2024], COVE [Wang et al., NeurIPS 2024], DistillDIFT [Fundel et al., WACV 2025]. In addition, standard optical trackers (CoTracker3, BootsTAP, ...) should also be compared to.
1. The claims made in l. 52..54 (also in other points of the paper, such as the abstract) are very general but not suffciently backed up by experiments: claims are made about flow extraction from generative model architectures in general, but the experiments throughout the paper only cover one family of approaches (propagating intervention points).
1. Many of the supposed advantages of "large world models" over classical methods proposed in 4.5 read more like limitations of pairwise flow estimation methods, not like limitations that would apply to optical tracking methods, which are also mainstream.
1. The proposed KL-based method can, out of the box, only be applied to a very limited range of generative video models and, importantly, does not directly apply to the prevalent model paradigm: diffusion models.
1. The checklist contains false claims, such as:
    a. 2: Limitations: not discussed in the section mentioned
    b. 7: Statistical significance: claimed to be reported, yet never provided

---

> ### Author Rebuttal · Authors · 2025-07-31
>
> We thank the reviewer for the thoughtful review and constructive suggestions! We address the main points below:
>
> **1. Diffusion Model Comparisons:** We appreciate the reviewer bringing up existing diffusion methods. We acknowledge that several important attempts have been made in this direction, and we will include this in our related works and discussions.
>
> After our submission, DiffTrack [1], a zero‑shot flow extraction procedure using a pre‑trained Diffusion Transformer, has been introduced. The authors include metrics from TAP‑Vid, and the method is shown to supersede many existing diffusion baselines, including DIFT [2]. The reviewer also brought up Track4Gen [3], which published comparable metrics that we present below (we note, however, that Track4Gen is **not** training‑free$^\dagger$, as their "zero-shot" evaluations are performed after they fine-tune the base model with trajectory annotations generated using optical flow).
>
>
> | TAP‑Vid DAVIS | AJ $\uparrow$ | $<\delta^x_{\text{avg}}\uparrow$ | OA $\uparrow$ |
> |:--------------|----:|---------------------------:|----:|
> | DIFT [2] | — | 39.7 | — |
> | Track4Gen$^\dagger$ [3] | 40.2 | 58.9 | **78.4** |
> | DiffTrack [1] | — | 46.9 | — |
> | KL‑LRAS (ours) | **44.2** | **65.2** | 74.6 |
>
>
> We see that on the commonly reported metric $<\delta^x_{\text{avg}}$, which measures the percent of points that fall within a set of error thresholds, our method outperforms SOTA diffusion baselines. Comparing with Track4Gen [3], we outperform or are competitive on all metrics, despite being completely training‑free.
>
> The reviewer also brought up additional concurrent work; however, at the moment direct comparison is challenging due to availability of code [4, 5]; difficulty in applying to non-generated videos (as mentioned in Sec. 6 of [6]); or differences in tasks [7], such as semantic correspondence vs. optical flow. Per the reviewer's suggestion, as public artifacts of previous methods become available, we will include more evaluations for comparison.
>
> In sum, many existing diffusion baselines have already been superseded by DiffTrack [1], against which our KL‑LRAS method shows improved results. By including the latest, highest-performing baselines, we hope to better contextualize our results. We will continue to incorporate additional comparisons as methods become publicly available.
>
> **2. Speed:** Extracting information from any large, generative video model is computationally demanding, as we discuss in Section 5. It just sort of "comes with the territory." However, it is still an important direction to take in order to leverage their deep understanding of scenes and motion. In fact, among the generative models we tested, our method is notably faster:
>
>    | Method | Relative time for point tracking |
>    |:-------|---------------------------------:|
>    | LRAS with KL‑tracing (ours) | **1×** |
>    | LRAS with RGB method | 8× |
>    | Cosmos | 68× |
>    | Stable Video Diffusion | 2x |
>
> But as discussed in our section on limitations, the natural next step to take after successful flow extraction—as is common in the literature on using general foundation models to build deployable, task-specific models—is to distill it into a faster architecture that can output dense flow in real time. The main purpose of our method therefore is to be used as a sparse *pseudo‑labeler*. This will significantly increase domain diversity for flow datasets, which have traditionally been limited to synthetic, or heavily curated scenes, without incurring the intractable cost of human labeling.
>
> Since submission, we have been actively exploring distillation strategies, where we too have been interested in understanding the effect of label sparsity. In particular, we have trained an existing, off‑the‑shelf dense flow model on a synthetic dataset with ground‑truth labels using different levels of sparsity. We find that a model trained on an extremely sparsified (0.03% of points labeled) dataset performs competitively to the same model trained on the original, dense dataset (endpoint error: 2.47 for dense, 2.63 for sparse), strongly encouraging us on the likely effectiveness of sparse distillation. Our goal is to have the distilled model ready for release with the NeurIPS camera-ready version, if accepted.
>
> **3. Exploration of different extraction procedures:**
>
> > Experiments throughout the paper only cover one family of approaches (propagating intervention points).
>
> To our knowledge, there are two main approaches for obtaining motion information from pre-trained models: (1) representation learning / mining approaches, as highlighted by the diffusion papers the reviewer mentioned, and (2) "propagating intervention points," as discussed in our paper. We acknowledge that our focus has primarily been on intervention-based approaches. This is because, if possible to execute, (2) is simpler, more generalizable, and naturally zero-shot. Also many other works have reported baselines on type (1) already — and which our results surpass.  But ultimately, we agree with the reviewer's suggestion to expand the discussion of other branches of related work by including results from the latest diffusion methods (DiffTrack, etc.) from the table above.
>
> **4. Support for Diffusion models:**
> > The proposed KL-based method ... does not directly apply to the prevalent model paradigm: diffusion models.
>
> This is an important point.  Though diffusion models are currently mainstream, and are great at generating photorealistic scenes, they have known difficulties with precise steering. Our results give a reason to continue exploring diffusion alternatives in future generative video models. (Other recent works provide independent reasons for this conclusion from different angles.) While methods that reinforce the use of a particular class of models that are currently popular can be valuable, we hope the same can be said for work that sheds light on complementary approaches.
>
> Our results expose two fairly general properties that, if present, are likely to be sufficient for good tracing: (1) locality, and (2) full distributional predictions. These criteria, while exposing a weakness of current diffusion models, are consistent with other classes of models that are actively being explored in the field. In fact, that diffusion models have this weakness is one of the points we are trying to make — one which we believe is quite valuable for the community.
>
> **5. Comparison with tracking methods:**
>
> > Limitations [do not] apply to optical tracking methods.
>
> To add context to this discussion, we present comparisons with CoTracker‑v3, the latest SOTA point tracker.
>
>    | TAP‑Vid DAVIS | AJ $\uparrow$ | AD $\downarrow$ | $<\delta^x_{\text{avg}}$ $\uparrow$ | OA $\uparrow$ |
>    |:--------------|----:|----:|---------------------------:|----:|
>    | CoTracker‑v3 [8] | 39.85 | 19.04 | 57.96 | **76.87** |
>    | KL‑LRAS (ours) | **44.16** | **11.18** | **65.20** | 74.58 |
>
> Point trackers face similar limitations, requiring synthetic data due to labeling difficulty (as discussed explicitly in [8]). The above results show CoTracker-v3 run under the same **two-frame** setting on TAP-Vid DAVIS to ensure fair comparison. The lower performance of CoTracker in this more difficult setting shows that the underlying motion estimation still remains challenging for optical trackers. Per the reviewer's advice, we will include this, and BootsTAP results with relevant discussion.
>
> **6. Limitations:** In the discussion, we do try honestly to mention the main limitation of our approach — the computational demand. We will further highlight this clearly in future versions.
>
> **7. Statistical Significance:** The TAP‑Vid benchmarks do not include standard procedures for measuring statistical significance, and most papers do not report such results. We interpreted the corresponding checklist entry to have been met by closely following the conventions of our domain, though we appreciate the reviewer bringing this to our attention. Part of the reason why this is not as emphasized in this domain is because the datasets are fairly large, so standard error bars tend to be extremely small and a little bit beside the point. Nevertheless, we computed the standard error for the Average Distance (endpoint error) metric for each model that we ran, as well as a paired t‑test to verify that our method significantly outperforms other models. For example, a one‑sided paired t‑test shows that our method achieves significantly lower endpoint error than Cosmos (p = 6e‑08). We will report these numbers in the supplement if they are believed to be important.
>
> [1] Nam et al., 2024. "Emergent Temporal Correspondences from Video Diffusion Transformers."
>
> [2] Tang et al., 2023. "Emergent Correspondence from Image Diffusion."
>
> [3] Jeong et al., 2025. "Track4Gen: Teaching Video Diffusion Models to Track Points Improves Video Generation."
>
> [4] Wang et al., 2024. "COVE: Unleashing the Diffusion Feature Correspondence for Consistent Video Editing."
>
> [5] Zhang et al., 2024. "Emerging Tracking from Video Diffusion."
>
> [6] Xiao et al., 2024. "Video Diffusion Models are Training-free Motion Interpreter and Controller."
>
> [7] Fundel et al., 2024. "Distillation of Diffusion Features for Semantic Correspondence."
>
> [8] Karaev et al., 2024. "CoTracker3: Simpler and Better Point Tracking by Pseudo-Labelling Real Videos."

---

> > ### Comment · Reviewer_XcXT · 2025-08-05
> >
> > I thank the authors for the extensive rebuttal and additional context & information provided.
> >
> > Some of my concerns have been resolved. I have some further questions below:
> >
> > > Speed: Extracting information from any large, generative video model is computationally demanding, as we discuss in Section 5. It just sort of "comes with the territory."
> >
> > I appreciate the comparison of relative speeds provided. Could the authors please also provide absolute speeds? It would also be important to contrast these speeds with other generative model-based tracking methods like the feature correlation-based approaches. Intuitively, I'd think that feature correlation-based approaches should be many orders of magnitude faster when predicting many tracks at once, which would mean that it does not necessarily "come with the territory".
> >
> > > Our results expose two fairly general properties that, if present, are likely to be sufficient for good tracing: (1) locality, and (2) full distributional predictions. These criteria, while exposing a weakness of current diffusion models, are consistent with other classes of models that are actively being explored in the field. In fact, that diffusion models have this weakness is one of the points we are trying to make — one which we believe is quite valuable for the community.
> >
> > This framing is better than the one implied in the paper but still not supported by the provided evidence in my opinion. Claiming that these two properties are *sufficient* for decent likelihood of getting good tracing capabilities is fine, but the second part of the claim - "that diffusion models have this weakness is one of the points we are trying to make" - does not follow from that. That would only be the case if the former were *necessary* conditions, which this work has not conclusively shown.
> >
> > I'd also urge the authors to address the concern raised by nvPE about the problems with comparison fairness due to the 2-view setting, which I share.

---

> ### Author Response · Authors · 2025-08-05
> **Specifically about the issue of "necessary" vs sufficient claims**
>
> The question about speed is important and I will let my colleague address that question separately.  Here I wanted to address the issue of the "necessary" vs "sufficient" claims.
>
> You're right that to totally mathematically tightly make the claim that diffusion methods could never under any circumstances do what our two "sufficient conditions" ask for would require a mathematical necessity proof.   But that kind of mathematical certainty seems usually unobtainable in empirical machine learning -- the problems are just too complicated for that kind of proof.
>
> Instead, what we can do is empirical studies that try to triangulate what is *likely* to be the case by gathering a bunch of evidence from different specific cases.  Here, what we've done is provide a clear framework (The two conditions) for what seems *likely* to be important for being able to do this type of zero-shot extraction from a generative model.  Our framework provides an understandable inductive reason why diffusion models might have trouble in this case -- because of the well-known issues they have with steerability -- we are far from the first to notice this problem, we just show evidence that this problem seems to affect the ability to do zero-shot optical flow.   This doesn't absolutely prove the case (of course), but it does give decent inductive evidence for it.
>
> It other words, we think it is likely that diffusion models have a weakness, and that if one wanted to solve the problem of zero-shot optical flow extraction from diffusion models, one would probably have to solve this weakness.   I don't think we think (or claim in our paper) that it's impossible to solve those issues, but just that they probably have to be dealt with if one wanted to use diffusion for this purpose.  Are you saying it would be better if we more thoroughly emphasized that this is an statement of "likelihoods" rather than "absolute certainties"?  We would be very happy to do that if it's not coming across strongly enough in the current write-up.
>
> Or are you saying a stronger thing -- that like maybe the inductive method of this kind shouldn't be deployed at all here? Like are you saying we should not make any suggestions at all about what seems likely (to us) to be true about diffusion until we have a air-tight proof of it somehow?  That type of conclusion would seem like it would rule out a lot of useful investigation in empirical machine learning ...?
>
> (Really sorry if we're not understanding your point yet... definitely not trying to give you a hard time, just trying to understand.)

---

> > ### Author Response · Authors · 2025-08-05
> > **making post visible to reviewer XcXT**
> >
> > Sorry, reviewer XcXT! Just wanted to make sure you saw the above comment, since by accident the first time we posted it, we might not have made is visible to you.

---

> > > ### Author Response · Authors · 2025-08-06
> > > **Regarding speed concerns of reviewer XcXT**
> > >
> > > Regarding speed, here are the absolute numbers for comparison that the reviewer requested (these numbers are on our old A40 hardware, and of course, as absolute timings, will change depending on the hardware type):
> > >
> > >
> > > | Method | Relative time for point tracking |
> > > |:-------|---------------------------------:|
> > > | LRAS with KL‑tracing (ours) | **3.3s/it** |
> > > | LRAS with RGB method | 28.2s/it |
> > > | Cosmos | 230.0s/it |
> > > | Stable Video Diffusion | 6.1s/it |
> > >
> > > Time comparisons for zero-shot extraction with other diffusion baselines besides SVD are difficult for the reasons we outlined in the rebuttal previously, but SVD is probably a decent guide for what other systems might achieve.
> > >
> > > Compared to methods that build on feature correspondence, zero-shot extraction from generative models by propagating interventions point-by-point is of course slower (like by a factor of 50-100x). But it is better, as our quantitative results show!  That's precisely because it gets at the "really correct" definition of flow -- namely, using the underlying world model to identify the way that one point in one frame causally determines a point in the next frame, rather than approximating this idea with imprecise heuristic assumptions about (e.g.) the smoothness of the correspondence, and thus enabling better results in complex domains. This is the rough tradeoff we are navigating: better results for more computation.
> > >
> > > But like we discussed at some length in the rebuttal, actually there is a way to nicely avoid this tradeoff: doing distillation with  pseudolabels from KL-LRAS (or whichever generative model one prefers).  From the results we described, it seems like this will end up being super effective and efficient, since almost perfect reconstruction of dense labels can be generated with very very sparse labels.  That this works very well is maybe not super super surprising to us since it has been observed for a while, kind of internally in the flow community, that fast feedforward flow extractors can be trained pretty sparsely -- and this fact is one of the reasons we've been ok with taking the zero-shot generative model approach. (Maybe this should be better explained in the paper?)
> > >
> > > Thus in practice, we will be able to achieve the best of both worlds -- actually really good quality results, extracted densely fast.  Does that seem like a reasonable approach to your mind?

---

> > > > ### Author Response · Authors · 2025-08-07
> > > > **Responding to last line of reviewer XcXT's response (re 2-view setting)**
> > > >
> > > > > I'd also urge the authors to address the concern raised by nvPE about the problems with comparison fairness due to the 2-view setting
> > > >
> > > > We have engaged in a more detailed discussion on this topic in response to reviewer nvPE, which we summarize below for convenience:
> > > >
> > > > ------
> > > >
> > > > One good way to think about this is: the 2-frame setting is the "core dynamics understanding" problem. It's harder than the multi-frame case. And thus all methods, whether they are co-tracker or KL-tracing or RAFT or whatever, will naturally post lower absolute numbers in the 2-frame setting as compared to the multi-frame setting, where it's possible to average over information from the multiple frames to boost the numbers. The 2-frame setting isn't "unfair", it's just different, and equally important, to the multi-frame setting.
> > > >
> > > > We've shown, with our 2-frame co-tracker comparison, that the supervised co-tracker method doesn't have some magical ingredient that makes it better at the base, hard optical flow case. (That's what we thought you would care about, since our whole paper is in the optical flow domain rather than tracking, to begin with.) In other words, we've shown that whatever higher numbers Cotracker is getting in the multi-frame case comes from the additional information of the multiple frames, not some deeply better dynamics-understanding principle. That's a fair comparison for us to make, and allows us to put that method on the same footing as all the other optical flow methods that we investigate in the paper (not just KL-tracing).
> > > >
> > > > To add to the discussion, we also ran evaluations for AllTracker [9] under a 2-frame constraint, as AllTracker can be viewed as being "more related to optical flow than CoTracker."  See results table below. For a more insightful comparison, we break down each result by frame gap (i.e. <= N indicates that only points with frame gap within N were evaluated), with "Full" being the full TAP-Vid First benchmark.
> > > >
> > > >   | TAP‑Vid DAVIS | AJ $\uparrow$ | $<\delta^x_{\text{avg}}\uparrow$ | OA $\uparrow$ |
> > > >   |:--------------|----:|---------------------------:|----:|
> > > >   | $\leq1$ | | | |
> > > >   | AllTracker | 67.6 | 72.8 | 99.2 |
> > > >   | KL-LRAS | 88.9 | 94.1 | 97.7  |
> > > >   | $\leq2$ | | | |
> > > >   | AllTracker | 58.8 | 65.9 | 98.9 |
> > > >   | KL-LRAS | 86.2 | 92.6 | 97.0 |
> > > >   | $\leq4$ | | | |
> > > >   | AllTracker | 48.3 | 56.7 | 96.8 |
> > > >   | KL-LRAS | 81.1 | 89.4 | 95.1 |
> > > >   | $\leq8$ | | | |
> > > >   | AllTracker | 35.9 | 45.0 | 92.5 |
> > > >   | KL-LRAS | 74.2 | 85.3 | 92.0 |
> > > >   | $\leq16$ | | | |
> > > >   | AllTracker | 25.6 | 34.3 | 88.1 |
> > > >   | KL-LRAS | 64.9 | 79.2 | 87.5 |
> > > >   | $\leq32$ | | | |
> > > >   | AllTracker | 18.4 | 26.7 | 82.1 |
> > > >   | KL-LRAS | 53.6 | 71.8 | 80.5 |
> > > >   | **Full** | | | |
> > > >   | AllTracker | 13.1 | 20.3 | 75.8 |
> > > >   | KL-LRAS | 44.2 | 65.2 | 74.6 |
> > > >
> > > > The AllTracker results below reveal a similar story to CoTracker: namely, multi-frame trackers tend to leverage information from the entire video, usually resulting in significantly improved numbers on these benchmarks compared to the 2-frame case. Even at short frame gaps (e.g., 1,2,4), the default AllTracker uses information from future frames of the video to perform tracking. This is why when run under a 2-frame constraint (table above), the performance degrades significantly for AllTracker. Again, not to say that one setting is "fairer" than the other, but just that they emphasize different things.
> > > >
> > > > The 2-frame constraint under which we evaluated all our baseline models directly evaluates a model's ability to understand the "core case" of challenging, real-world dynamics. 2-frame optical flow is of course useful for a bunch of applications, and performance numbers like these speak to the utility of algorithms (such as KL-LRAS) in those applications. There is also a decent chance that 2-frame improvements here can transfer to the multi-frame domain and result in an equally improved tracker, though showing that will have to be a subject for future work...
> > > >
> > > > [9] Harley et al., 2025. "AllTracker: Efficient Dense Point Tracking at High Resolution."

---

> > > > > ### Comment · Reviewer_XcXT · 2025-08-08
> > > > >
> > > > > I thank the authors for the additional clarifications and evaluations.
> > > > >
> > > > > My main concern about the extremely low speed, which has been confirmed to be many orders of magnitude lower than existing methods (e.g., the SOTA method TapNext manages 40+k tracks * frames / second on a V100 vs. 0.3 tracks * frames / second for the proposed method on an A40, with both GPUs being similarly powerful), remains. Similar concerns apply to SOTA dense optical flow estimation methods.
> > > > >
> > > > > Regarding W2, I consider that point reasonably resolved.
> > > > >
> > > > > I do not consider the additional material provided during the rebuttal sufficient to justify the broad statements made in the original submission (W3 & W4). I expect the authors to address that as discussed and match claims made to the evidence provided in the next revision.
> > > > >
> > > > > Under the assumption that W2-4 will be properly addressed in the next revision, I will adjust my rating.

---

### Official Review · Reviewer_uSVM · 2025-07-03

**Clarity:** 2
**Significance:** 3
**Originality:** 3
**Rating:** 4
**Confidence:** 3

**Summary:**

This paper proposes KL-tracing, a novel zero-shot inference approach for extracting optical flow from pretrained generative world models. Inspired by the concept of counterfactual world models (CWMs), the authors introduce a perturb-and-track procedure based on KL-divergence in latent spaces, tracing subtle perturbations through predictions of future frames. Evaluations conducted on the challenging TAP-Vid DAVIS and Kubric benchmarks demonstrate convincing performance improvements, especially when using the LRAS generative model.

**Questions:**

1. Could the authors discuss more concretely what would be required to make KL-tracing work with architectures beyond LRAS? The performance on cosmos and SVD is not strong.

2. How sensitive is KL-tracing to the magnitude and shape of the perturbation? Would a different form (e.g., a low-frequency mask or structured noise) work similarly?

**Ethical Concerns:**

["NO or VERY MINOR ethics concerns only"]

**Final Justification:**

Most of my concerns have been addressed. However, I still think the proposed method is restricted to the certain type of model, i.e., LRAS, which limits the applicability of the proposed to broader families of video models.

Overall, this is an interesting paper, but there is still room for improvement in terms of quality. I will maintain my current rating for now (positive recommendation) as an encouragement.

**Limitations:**

yes

**Quality:**

3

**Strengths And Weaknesses:**

Strengths:
1. The idea in this paper is interesting, although it takes time to digest due to the writing clarity. The use of KL-divergence in latent distributions to trace perturbations and the perspective of counterfactual world models are not commonly seen in this direction. It feels novel and offers a fresh take on optical flow to me as a reviewer.

2. The empirical results on the TAP-Vid benchmarks are good and convincing.

3. Despite could-be-improved writing clarity, the authors do a commendable job walking through how the counterfactual idea applies to different types of generative models (e.g., LRAS, Stable Video Diffusion, Cosmos). As a reviewer, I appreciate the effort to bridge the conceptual gap across these architectures.


Weakness:
1. The best performing setup relies on a specific model (LRAS), which somewhat limits the broader applicability of the method. The authors provide analysis for SVD and Cosmos, but those results are weak. I believe the evaluations of these two models should also be included in Table 2 for completeness, as that table is the clearest comparison with traditional flow baselines.

2. The writing could be greatly improved. The introduction assumes the reader is already familiar with Counterfactual World Models, which may not be true for everyone. Also, the paper emphasizes LRAS architecture too heavily, despite it not being the core contribution here. Lastly, I’d suggest reorganizing the method sections more cleanly—there’s noticeable redundancy (e.g., L117–L125 repeat in L180–L191 almost verbatim).

I feel the paper would benefit from a few rounds of polishing and iteration to reach a more refined state. That said, I like the core idea and I encourage the authors to keep improving this work.

---

> ### Author Rebuttal · Authors · 2025-07-31
>
> We thank the reviewer for the thoughtful review and constructive suggestions! Overall, we are glad that reviewers found our method "novel" and "interesting" (uSVM) and our empirical results and analyses to be "strong" and "detailed," and a significant improvement to existing methods (fTmd, uSVM, nvPE). We address the main points of the reviewer below:
>
> > The best performing setup relies on a specific model (LRAS), which somewhat limits the broader applicability of the method.
>
> > Could the authors discuss more concretely what would be required to make KL-tracing work with architectures beyond LRAS?
>
> The key properties needed to make KL tracing work are (1) locality (local tokenizer enables detailed perturbation control) and (2) distributional prediction (predicting entire distributions instead of a single sample). Please refer to Section 3.5 for more detail. These features are model-agnostic and can be incorporated into a generative model's architecture in various ways. While the mainstream models of today (e.g., Cosmos, SVD) may lack these properties, and among existing models LRAS happens to have them, we believe that our results highlight an alternative direction for building powerful future generative video models, since the requirements that KL tracing suggest are not overly restrictive. For clarity, we will include further discussions outlining the general properties we have identified to work particularly well for KL tracing.
>
> > I believe the evaluations these two models should also be included in Table 2 for completeness.
>
> We agree that placing SVD and Cosmos results directly in Table 2 makes the comparison clearer, and will add full results for both models in the revision. At the moment, to help with the discussion and contextualize diffusion-based extraction, we include TAP-Vid results from DiffTrack [1], a recent SOTA paper that uses a model-specific readout on Diffusion Transformers (DiT) for zero-shot optical flow:
>
> | TAP‑Vid DAVIS                 | $<\delta^x_\text{avg}\uparrow$   |
> |:-----------------------------|-----------------:|
> | SEA‑RAFT                     | 58.7 |
> | DiffTrack [1] |        46.9 |
> | KL‑LRAS (ours)               |     **65.2** |
>
> This shows that flow extraction from diffusion models remains challenging compared to KL tracing used with LRAS.
>
> > How sensitive is KL-tracing to the magnitude and shape of the perturbation? Would a different form (e.g., a low-frequency mask or structured noise) work similarly?
>
> We thank the reviewer for engaging with our work and proposing interesting ablations. We evaluated this using the same setting as in Table 1. For "shape" and "magnitude", we vary different parameters of our Gaussian bump perturbation such as amplitude ($\alpha$) and standard deviation ($\sigma$). For the low-frequency mask perturbation, we use a standard Gaussian blur kernel. We find empirically that our choice of a Gaussian "bump" perturbation is quite important, though the results are less sensitive to the specific parameters. However, we are open to the idea that a more focused investigation on the optimal perturbation may yield better results or a different form. Our results are below:
>
>  | Pert Type | $\alpha$ | $\sigma$ |  EPE $\downarrow$ |
> |:----------|----------:|--:|-----:|
>  | Gaussian  |      -200 | 2 | 5.30 |
> | Gaussian  |      -100 | 2 | 4.90 |
>  | Gaussian  |       100 | 2 | 5.45 |
> | Gaussian  |       200 | 2 | 5.10 |
> | Gaussian  |       255 | 1 | 6.45 |
> | Gaussian  |       255 | 3 | 5.21 |
> | Gaussian  |       255 | 4 | 6.55 |
> | Gaussian  |       255 | 5 | 7.05 |
> | Low Freq. Mask |       255 | 2 | 41.61 |
> | **(baseline) Gaussian**        |       255 | 2 |  5.08 |
>
> Finally, we greatly appreciate the concerns brought up about writing clarity. We will take the reviewer's advice and (1) include a more proper introduction of Counterfactual World Models, (2) do a heavy reorganization of the method section and (3) add more discussions directly related to our contributions.
>
> [1] Nam et al., 2025. "Emergent Temporal Correspondences from Video Diffusion Transformers."

---

> > ### Comment · Reviewer_uSVM · 2025-08-05
> >
> > I thank the authors for their detailed response. Most of my concerns have been addressed. I understand that there may not have been sufficient time to include the results of SVD and Cosmos in Table 2, but I hope the authors will consider adding these results in the future, as they would arguably provide a fairer comparison than directly quoting numbers from existing works.
> >
> > Overall, this is an interesting paper, but there is still room for improvement in terms of quality. I will maintain my current rating for now (positive recommendation).

---

> > > ### Author Response · Authors · 2025-08-05
> > > **important clarification**
> > >
> > > We thank the reviewer for considering our rebuttal. We would like to make one **important clarification**. The numbers reported for Cosmos and SVD were not quoted from another paper but in fact results from experiments which we ourselves ran after investing significant engineering effort to implement them. They are reported on a smaller validation set but we also offer LRAS numbers for comparison. Cosmos and SVD on the smaller validation set show they are worse than KL-LRAS (ours) by a large margin (KL-LRAS: 5.0762, SVD: 74.7990, Cosmos: 35.4338) (lower is better) and we believe the relative trends would likely hold for the full TapVid dataset. Cosmos and SVD take significantly longer to run than our method, so it was impractical to run on the full dataset during the rebuttal period, but we can do it for the final version. Furthermore the very reason why we developed this validation set is because we want to try several different methods for probing these models instead of just running the most naive implementation (which yielded significantly worse results) on the full dataset. **We feel we really performed a rigorous analysis of these models and so want to clarify any misunderstanding (see section 3.3 and 3.4 of our original submission for details).**

---

### Official Review · Reviewer_fTmd · 2025-07-04

**Clarity:** 3
**Significance:** 3
**Originality:** 3
**Rating:** 4
**Confidence:** 3

**Summary:**

This paper investigates the potential of extracting optical flow in a zero-shot manner from frozen, large generative video world models. Motivated by the limitations of prior supervised and photometric approaches, the authors systematically evaluate deterministic, diffusion, and autoregressive generative models. They show that successful flow extraction requires distributional prediction, factorized local latents, and random-access decoding. The paper introduces KL-tracing, a test-time inference procedure that measures the Kullback-Leibler divergence between clean and perturbed model predictions to robustly localize motion and extract flow without model fine-tuning. Experimental results on real-world (TAP-Vid DAVIS) and synthetic (TAP-Vid Kubric) datasets demonstrate that KL-tracing with LRAS surpasses both classical and state-of-the-art flow models, even though the world models were not trained with flow supervision.

**Questions:**

The idea is now new, as it has been proposed in papers like SiamMAE and other papers. The authors are encourage to do more experiments including both ablation and comparison with other SOTA methods.

**Ethical Concerns:**

["NO or VERY MINOR ethics concerns only"]

**Final Justification:**

The author addressed all my concerns in the rebuttal, I inclined to accept the paper

**Quality:**

3

**Strengths And Weaknesses:**

The paper presents detailed and comparative results across various generative world models (deterministic, diffusion, autoregressive, and LRAS), using standard real-world and synthetic benchmarks. The results, particularly in Table 2, demonstrate strong empirical performance, with KL-LRAS outperforming both supervised and unsupervised baselines in several key metrics.

Despite strong performance on TAP-Vid, the paper does not empirically demonstrate zero-shot transfer to significantly out-of-distribution domains, such as medical, aerial, or night-time video. While the theoretical basis for generalization is sound, more experimental evidence or at least critical discussion would be helpful. In addition, although the performance is good as shown in the Figure, the paper failed to compare with other tracking methods like cotracker3. We encourage the authors to do more compareison with other SOTA optical flow or tracking methods.

---

> ### Author Rebuttal · Authors · 2025-07-31
>
> We thank the reviewer for the thoughtful review and constructive suggestions! Overall, we are glad that reviewers found our method "novel" and "interesting" (uSVM) and our empirical results and analyses to be "strong" and "detailed," and a significant improvement to existing methods (fTmd, uSVM, nvPE). We address the main points of the reviewer below, highlighting transferability and additional tracking baselines:
>
> > The paper does not empirically demonstrate zero-shot transfer to significantly out-of-distribution domains.
>
> The reviewer brings up an interesting point of transferability and generalizability. We believe that this is demonstrated, at least to some extent, in our work both quantitatively and qualitatively. Although the base LRAS model is trained on real-world videos, KL-LRAS, with no additional training, performs strongly on both real (TAP-Vid DAVIS) *and synthetic* (TAP-Vid Kubric) benchmarks (Table 2), demonstrating strong cross-domain generalization within the larger range of "normal" videos containing people and objects. Further, qualitative results in Figures 1, 6, and the attached video demo also show robustness on challenging, long-tail motions (e.g., motion blur or in-place rotation).
>
> To be honest, however, we don't know if a model trained on YouTube videos will transfer to very different kinds of videos where notions of object and timescale are extremely different (like medical data). Regardless, our approach, KL tracing, works with any base predictor that is effective at its underlying prediction task, whatever the domain may be. Thus—and this is the most important point—the main advantage of our approach is that we can easily extend our method to handle any new domain merely by extending the training data of the base predictor model of which our approach operates on top. We don't need to get new labels for that domain, bypassing the intractable cost of obtaining human labels for anything other than synthetic, carefully curated computer-graphic-type scenes. As foundation models improve across domains, our method's performance scales accordingly without requiring domain-specific flow annotations.
>
> We will make this point clearer, and also include further qualitative examples of out-of-domain "stress tests" such as performance on night-time videos (please note that we are unable to submit additional demos as part of this rebuttal).
>
> > The paper failed to compare with other tracking methods like cotracker3
>
> We agree that this reference is informative and therefore include results for CoTracker-v3.
>
> | TAP-Vid DAVIS                           | AJ $\uparrow$  | AD $\downarrow$ | $<\delta^x_\text{avg}\uparrow$ | OA $\uparrow$   |
> |:----------------------------------------|-------:|-------:|------------:|-------:|
> | CoTracker‑v3 [1]      | 39.85  | 19.04  | 57.96       | **76.87** |
> | KL‑LRAS (ours)                          | **44.16** | **11.18** | **65.20** | 74.58 |
>
>
> We note that the above results show CoTracker-v3 run under the same **two-frame** setting on TAP-Vid DAVIS to ensure fair comparisons with the flow and generative model approaches. The lower performance of CoTracker in this more difficult setting shows that the underlying motion estimation still remains challenging for trackers. Following the reviewer's advice, we will also add additional point tracker and flow baselines to our paper, as well as express the above point more clearly.
>
> [1] Karaev et al., 2024. "CoTracker3: Simpler and Better Point Tracking by Pseudo-Labelling Real Videos."

---

> > ### Comment · Reviewer_fTmd · 2025-08-08
> >
> > The paper again demonstrates the potential of generative models for solving video-as-input tasks, similar to other works using generative video models like Aether. We encourage authors to discuss these related works and add experiments of cotrackerv3 in the final version. I inclined to maintain the original score.

---

### Comment · Area_Chair_YZqH · 2025-08-04
**Response required.**

Dear reviewers,

As the Author–Reviewer discussion period concludes in a few days, we kindly urge you to read the authors’ rebuttal and respond as soon as possible.

- Please review all author responses and other reviews carefully, and engage in open and constructive dialogue with the authors.

- The authors have addressed comments from all reviewers, but only one reviewer has responded so far; each reviewer is expected to respond, so the authors know their rebuttal has been considered.

- We strongly encourage you to post your initial response promptly to allow time for meaningful back-and-forth discussion.

Thank you for your collaboration,
AC

---

> ### Comment · Area_Chair_YZqH · 2025-08-05
> **Reviewer Participation in Discussion Period**
>
> Dear Reviewers,
>
> ACs have been instructed to flag non‑participating reviewers using the Insufficient Review button (Reviewers fTmd and uSVM).
> Please ensure that you read the rebuttals and actively contribute to the discussion.
>
> Kindly note that clicking Mandatory Acknowledgement before the discussion period ends does not exempt you from participating. This should only be submitted after you have read the rebuttal and engaged in the discussion.
>
> Best regards,
> AC

---

### Note · Authors · 2025-08-14

Thank you to the AC for moderating the discussion and to all reviewers for the thoughtful feedback! We greatly appreciate that reviewers found our work to be "novel," "interesting," "strong," and "detailed" – we highlight a couple of key points below:

**Additional comparisons (fTmd, uSVM, XcXT, nvPE):** We added the latest trackers: CoTracker and AllTracker. KL-LRAS (ours) outperforms both in the two-frame setting, which directly tests a models' "core dynamics understanding." We discuss the merits of the two-frame evaluation in detail with reviewers nvPE and XcXT – we are glad that it seems that reviewers are largely aligned here. We also added results for DiffTrack, Track4Gen, and DIFT, as suggested by reviewer XcXT, against which KL-LRAS remains notably strong.

**Speed (XcXT, nvPE):** We reported runtimes across generative baselines. While per-point KL-tracing can be slower than feature-correspondence mining, it delivers superior accuracy and is faster than other generative readouts we evaluated. In practice, we can use KL-LRAS as a pseudo-labeler for sparse distillation into a dense-flow model, mitigating the speed/quality trade-off. We discussed this in greater detail with reviewer XcXT.

We also addressed other questions, e.g., ablations on perturbation size/type (uSVM); OOD transferability (fTmd); and NeurIPS checklist clarifications (XcXT).

Concretely, for revisions, we will:
 * Make the suggested writing changes, e.g., reorganize methods section (uSVM), and moderate our language to better scope our results (XcXT)
 * Add Cosmos and SVD numbers to the full TAP-Vid benchmark table (Table 2)
 * Include trackers and diffusion models in related works and result comparisons
 * Present a thorough discussion on speed/quality-trade off and distillation process/results

And, of course, we will incorporate other actionable items brought up. Overall, we're grateful for the dialogue and appreciate that reviewers raised or maintained their positive scores accordingly!

---

### Decision · Program_Chairs · 2025-09-17

**Decision:**

Accept (poster)

**Comment:**

This paper introduces KL-tracing, a prompting-based approach for extracting optical flow from frozen self-supervised video world models. Built on the LRAS architecture, the method achieves state-of-the-art results on TAP-Vid DAVIS and Kubric, offering a novel and conceptually interesting counterfactual perspective on flow estimation without fine-tuning. The simplicity of the approach and its strong empirical performance on challenging benchmarks make it a notable contribution.

While reviewers raised concerns regarding the limited applicability of the method (with strongest results on LRAS), lack of evaluation on standard optical flow datasets, and computational efficiency, the rebuttal addressed several points satisfactorily. Importantly, reviewers recognized the novelty and potential of the idea, with two maintaining positive recommendations and one increasing their score to Accept. Only one reviewer continued to lean toward rejection, primarily due to concerns about generality and speed.

On balance, the strengths in novelty, conceptual clarity, and empirical performance outweigh the limitations. The paper can be accepted.